# Capillary Wicking on *Heliamphora minor*-Mimicking Mesoscopic Trichomes Array

**DOI:** 10.3390/biomimetics9020102

**Published:** 2024-02-09

**Authors:** Fenglin Chen, Ziyang Cheng, Lei Jiang, Zhichao Dong

**Affiliations:** 1CAS Key Laboratory of Bio-Inspired Materials and Interfacial Science, Technical Institute of Physics and Chemistry, Chinese Academy of Sciences, Beijing 100190, China; 2School of Future Technology, University of Chinese Academy of Sciences, Beijing 100049, China

**Keywords:** capillary wicking, *Heliamphora minor*, 3D printing, mesoscopic trichomes

## Abstract

Liquid spontaneously spreads on rough lyophilic surfaces, and this is driven by capillarity and defined as capillary wicking. Extensive studies on microtextured surfaces have been applied to microfluidics and their corresponding manufacturing. However, the imbibition at mesoscale roughness has seldom been studied due to lacking fabrication techniques. Inspired by the South American pitcher plant *Heliamphora minor*, which wicks water on its pubescent inside wall for lubrication and drainage, we implemented 3D printing to fabricate a mimetic mesoscopic trichomes array and investigated the high-flux capillary wicking process. Unlike a uniformly thick climbing film on a microtextured surface, the interval filling of millimeter-long and submillimeter-pitched trichomes creates a film of non-uniform thickness. Different from the viscous dissipation that dominated the spreading on microtextured surfaces, we unveiled an inertia-dominated transition regime with mesoscopic wicking dynamics and constructed a scaling law such that the height grows to 2/3 the power of time for various conditions. Finally, we examined the mass transportation inside the non-uniformly thick film, mimicking a plant nutrition supply method, and realized an open system siphon in the film, with the flux saturation condition experimentally determined. This work explores capillary wicking in mesoscopic structures and has potential applications in the design of low-cost high-flux open fluidics.

## 1. Introduction

Liquid touching superlyophilic rough surfaces spontaneously spreads into the texture, driven by capillarity; this is called capillary wicking [1,2,3,4,5,6,7] and is of significant interest in a broad range of applications such as microfluidics [8,9,10,11], the design of self-cleaning surfaces [12], and lab-on-a-chip technology [13,14,15]. Extensive research on the wetting of microtextured surfaces [1,2,3,4,5,6,10,16,17,18,19,20,21,22,23] demonstrates that liquid behavior on surfaces with microscopic structures is influenced by the combined effects of the surface wetting properties, structure shapes, and liquid properties. In detail, surface energy serves as primary driving force, while viscous dissipation hinders the spreading or imbibition of liquid [24,25], and the surface roughness, defined as the ratio between the actual and projected surface areas, enhances the release of surface energy [1,2,26]. The scaling law known as the Lucas–Washburn equation [27] states that the macroscopic wicking distance increases to 1/2 the power of time due to the balance between surface tension and viscosity. The initial stage is dominated by inertia [28], which follows a power law of 1. Furthermore, a microscopic view of a fringe film extending into the roughness interval was addressed recently, in which the meniscus grew to 1/3 the power when the front was far from the microscale structure [4,29]. However, microtextured surfaces cannot achieve high-flux wicking due to the limitation posed by their structure size and film thickness. To utilize spontaneous capillary wicking in practical applications, it is essential to explore and understand the dynamics of wetting the modified surfaces of large-scale structures.

Inspired by the plant strategy of modifying the surface with millimeter-long trichomes to manipulate their interaction with water for survival, we systematically investigated the capillary wicking on *Heliamphora minor*, a representative species of South American pitcher plants, and revealed that a lubrication layer spontaneously forms on the pubescent wall to capture insects as well as assist in draining excess water from the pitcher [30,31]. With the speed and resolution improvement of additive manufacturing, the rapid preparation of mesoscale structures is convenient, allowing for the fabrication of a mimetic millimeter-long trichomes array for further study of the wicking dynamics under various experimental conditions. We discovered that the wicking on mesoscopic trichomes forms a non-uniformly thick film which enables high flux flow and explained the profile of this film with respect to the concept of the catenary line. Moreover, we demonstrated that the wicking dynamics on mesoscopic structures differs from that on microscopic structures, with the distance growing to 2/3 the power of time. Additionally, we used the mimetic trichomes array to construct an open siphon and examined the change of flux with height. Through a simulation using salt and sugar, we demonstrated the mass transportation in a thick wicking film, displaying an active method utilized by plants for nutrient supply. Overall, this work provides innovative insight into the design of mesoscopic structures that achieve spontaneous capillary wicking, and advances our understanding of this phenomenon for practical applications requiring high flux.

## 2. Materials and Methods

### 2.1. Biological Characterization and the Fabrication of Biomimetic Samples 

*Heliamphora minor*, whose lid is smaller than the pitcher (Figure 1a, images taken by Nikon D750, Toyko, Japan), was cultivated in a greenhouse with the environment maintained at a temperature of 23–28 °C, and a humidity of 90–95%. The sectional view from the middle (Figure 1b) presents the pitcher’s inside wall morphology, with a constriction that splits the wall into two zones; the upper is covered with millimeter-long trichomes and called the pubescent zone, and the lower is the smooth zone. A stereomicroscope (Figure 1c, DSX 1000, Olympus, Toyko, Japan) and scanning electron microscopy at 10 kV (Figure 1d, SEM, SU8020, Hitachi, Toyko, Japan) were utilized for detailed characterization of trichomes on the inside wall (Figure 1e for statistical analysis by Origin 2024). The trichomes are downward-pointing and densely arranged, with a base diameter, *d*, of 49.3 ± 18.1 μm, and a mutual pitch, *p* = 176.1 ± 36.9 μm. The trichomes’ length, *L*, ranges from 153.4 μm to 922.2 μm, with an average of 547.1 μm. In general, the trichomes’ distribution is in mesoscale, which is ubiquitous in nature [32].

Contact angle measurements of the pubescent zone (Figure 1f–h, DCAT 21, DataPhysics, Filderstadt, Germany) indicate that the wettability is superhydrophilic and water spontaneously wicks into the trichomes as soon as it touches the surface. The lightning-fast wicking process was observed and recorded using a high-speed camera at 1000 fps (Figure 1i,j, FASTCAM Mini UX100, Photron, Toyko, Japan); water injected by a syringe from a distribution pump (TS-1A, Longer Pump, Baoding, China) at a rate of 200 μL s^−1^ spread radially on the pubescent wall and filled the interval at the same time. Therefore, the film thickness increases as the height becomes higher. Finally, the film covers the trichomes and serves as the lubrication layer. Mimicking the pubescent zone, a polymerization-based 3D printer (Mini 8K, Phrozen Tech., Taiwan, China) using an ultraviolet liquid crystal display to initiate the photo reaction of the monomer methyl methacrylate—TiO_2_–SiO_2_ mixture was used (Figure 1k), and the layer resolution and pixel resolution were 20 μm and 18 μm, respectively. After preparation, the mimetic samples were ultrasonically washed in commercial cleaning solutions (Go Print Co., Ningbo, China) for 5 min, dried by N_2_ flow and post-cured under UV light for 3 min. Oxygen plasma treatment (DT-03, OPS Plasma Technology Co., Shenzhen, China) at an RF power of 300 W for 10 min was performed to adjust surface superhydrophilicity. SEM images of the printed mimetic samples sputtered with a thin layer of platinum (EM ACE, Leica, Wetzlar, Germany) at 10 kV were obtained (Figure 1l). The particles of TiO_2_ and SiO_2_ increase surface roughness and enable chemical modification (Figure 1m for energy-dispersive spectrometer).

### 2.2. Capillary Wicking, High-Flux Siphon Application, and Mass Transportation

The fabricated biomimetic trichomes array was inserted into a dish filled with blue-dyed water (Figure 2a as a schematic), and the dish diameter was 10 cm, which was large enough to avoid the wicking process disturbing the surface. A colored 4K resolution high-speed camera at 1000 fps (Wave, Freefly, WA, USA) was used to observe and record the wicking dynamics from the side, front, and oblique view. The water fringe front extending on substrates of various trichome geometries was tracked by Tracker software (6.0.10 version), starting from the moment of contact with the water level. Additionally, we utilized micro-computed tomography (Micro-CT, Skyscan 1272, Bruker, German) to reconstruct the wicking film profile and investigate the formed curvature gradient. Furthermore, mimicking the biological function of assisting drainage from pitcher, we modified an n-shaped arc surface of mesoscale trichomes, which was also prepared by 3D printing, and put the higher side of the model in contact with water tank, measuring the mass of water flowing out using an analytical balance (104E, Mettler Toledo, Switzerland). The flux was reported in the form of the measured mass divided by time. Finally, we dissolved CuSO_4_, FeCl_3_, and glucose, respectively, into water and repeated the wicking process. After long enough time for evaporation, the solute crystallization from the solution process was recorded using a Nikon D750, which reveals the mass transportation inside the wicking film.

## 3. Results and Discussion

### 3.1. Capillary Wicking Height and Film Thickness

Once water touched the mesoscopic trichomes array, wicking between the trichomes was observed on the hydrophilic and superhydrophilic substrates (Appendix A) whose contact angles were smaller than the hydrophilic–hydrophobic boundary of 65° [33]. Conversely, water did not climb on the hydrophobic substrate. The front of the advancing meniscus exhibits a parabolic shape (Figure 2b) until it is fully developed at approximately 50 ms. Then, water climbing on the substrate and filling the intervals of the trichomes occur at the same time, which results in a film of non-uniform thickness and only wets the surface of the trichomes at higher heights (Figure 2c).

Similar to the capillary rise in a tube with a small radius (Appendix A), where the equilibrium rising height decreases as the diameter increases, known as Jurin’s law [34], the equilibrium wicking height, *H*, decreases as the pitch increases (Figure 2d). Notably, the mesoscopic structures of the discontinuous periodical biomimetic trichomes differ from the continuous tube wall. On the former substrate, the meniscus must extend from the first row to reach the second row (Figure 2e, inset) to ensure a continued wicking process. The pitch limitation, *p*_m_, of our fabricated material was determined to be 2200 µm, above which no obvious capillary wicking and interval filling were observed. This limitation is attributed to an insufficient driven pressure in the first interval, while the maximum pitch of two closet trichomes satisfies the condition where the pressure at the advancing meniscus in the first interval, *P*_M_, is balanced by the hydrostatic pressure, *P*_h_, such that
*P*_M_ = *P*_air_ − *σL*/*p*_m_^2^ = *P*_h_ = *P*_air_ − *ρ*g*p*_m_(1)
where *σ* is liquid–air surface tension, *ρ* is the density of the liquid, and g is the acceleration of gravity. The maximum pitch obtained from Equation (1) is *p*_m_ = (*Ll*_c_)^1/3^, where *l*_c_ is (*σ*/*ρ*g)^1/2^ and called the capillary length. For the *L* = 2 mm used in our experiments, Equation (1) predicted a pitch limitation of 2.4 mm which is consistent with our results.

In addition, the film thickness in single interval changes with time, not monotonically increasing but oscillating around the final thickness. This oscillation starts after the water imbibes into the interval and lasts for O(0.1 s), which is particularly evident in intervals at lower heights, indicating that previously filled intervals act as a source to supply the meniscus extending to the next row and repeating the filling process.

### 3.2. Microscopic Capillary Wicking Dynamics

An enlarged observation of the capillary wicking process (Figure 3a) shows that the advancing meniscus front (red dotted line) extends over the substrate at the same velocity, but water first wets only the bottom of trichomes. Driven by capillary forces, *F*_c_, along the perimeter of trichomes (Figure 3b), water gradually fills the interval and eventually covers almost the whole surface. The competition between the extension of the advancing meniscus on the substrate and the water spreading on the surface of trichomes contributes to the oscillation of film thickness in single interval that delays the spreading process.

Racing to the *n*th interval from the bottom to the top, the capillary forces, *F*_c_ = *σπd_n_*, where *π* is the circular constant and *d_n_* is the contact diameter of the trichomes in the *n*th row, decrease as the water spreads towards the tip of the trichomes due to their smaller *d_n_*. Additionally, the film thickness in the *n*th interval, *e_n_*, is non-uniform and decreases as the row number, *n*, increases (Figure 2c). Consequently, the capillary force along the perimeter of the trichomes varies with *n* and deforms the liquid–air interface in each interval to different extent during the filling process. Because *e_n_* is obviously influenced by the trichome length, *L*, as longer trichomes support a thicker film, the ratio between *e_n_* and *L,* denoted as *α_n_*, is utilized as a dimensionless quantity to describe the profile of the film. In the case of trichome lengths of 1 mm, 2 mm, and 3 mm, the variation of *α_n_* with *n* shows a similar tendency, suggesting a self-similarity of the film profile.

To investigate the relationship between the thickness and the meniscus profile, we employ the catenary line concept for detailed analysis and description (Figure 3d). As shown in the left schematic, we consider an arbitrary arc of a small length, *PQ*, in the *n*th interval, and the difference in the z direction between points *P* and *Q* is denoted as ∆z. The force balances between the surface tension force, *T*(z), and gravity, *G*(z), in the horizontal and vertical directions are
*T*(z) sin(*θ*(z)) = *T*(z + ∆z) sin(*θ*(z + ∆z)),(2)
*T*(z + ∆z) cos(*θ*(z + ∆z)) − *T*(z) cos(θ(z)) = *G*(z) = *ρ*g*S_n_*[*φ*(z + ∆z) − *φ*(z)](3)
where *φ*(z) is the path of the profile from point B to P, *θ*(z) is the tangential angle with respect to the z axis, and *S_n_*~*e_n_p* is the cross-sectional area perpendicular to the z axis. Considering an infinitely small step as ∆z approaches 0, and dividing Equations (2) and (3) by ∆z, their differential forms are given as
*d*(*T*sin*θ*)/*d*z = 0,(4)
*d*(*T*cos*θ*)/*d*z = *ρ*g*S_n_dφ*/*d*z(5)
Equation (4) implies that *T*sin*θ* = *F,* where *F* is a constant comparable to the surface tension force along the contact perimeter of the trichomes, which scales as *σd_n_*. Substituting *T* into Equation (5) as well as the path, *φ*, by the profile function, *f*(z), obtains an equation describing the *f*(z):−*f*″/(*f*′)^2^ = *K_n_* (1 + (*f*′)^2^)^1/2^(6)
where *f*′ and *f*″ denote the first and second derivatives of the profile, *f*, and *K_n_* = *ρ*g*S_n_*/*F* is a constant related to the film thickness, *e_n_*. The boundary conditions are rigorously adjusted by the curvature at the contact points *A* and *B* to satisfy the Laplace pressure balance of the hydrostatic pressure.

The general solution of Equation (6) describes the meniscus profile in the *n*th interval (see Appendix B for the mathematical derivation). To visualize the profile, we plot normalized z coordinates, z* = z/*p*, against normalized y coordinates, y* = y/*L*, for different film thickness ratios *α* (Figure 3e). As the film thickness in the interval decreases, the meniscus becomes flatter and the curvature of the meniscus, *κ*, induced by the deformation of the profile, *f*, is
*κ* = −*K_n_*/(*K_n_*z + *C*)^2^(7)
where *C* is a negative constant determined by the boundary condition of the profile. As indicated by Equation (7), the curvature-determined Laplace pressure increases from the bottom to the top of each interval (z increases from 0 to *p*). In other words, a pressure gradient forms inside the film at the *n*th interval, which propels the meniscus front to extend on substrate to reach the (*n* + 1)th row of trichomes. As the meniscus advances, the capillary force along the perimeter of the trichomes drives the liquid spreading along the trichomes to fill the (*n* + 1)th interval and finally renews the pressure gradient, with the coefficient *K_n_* modified to *K_n_*_+1_.

The repeated behavior of substrate climbing and interval filling results in different meniscus profiles in each interval along the wicking direction (Figure 3f). Cross-sectional views perpendicular to the z axis and in the y–z plane prove that this pressure gradient; that the pressure at the bottom is large, and small at the top. In a single interval (Figure 3g), the release of the trichomes’ surface energy accelerates the flow, as water reaches the base of the trichomes while viscous dissipation slows down the flow, which is consistent with previous work on the wetting of rough surfaces decorated with sparsely arranged microscopic structures [4]. In general, the capillary force acting on the perimeter of a trichome leads to a catenary line-shaped meniscus profile, which induces a pressure gradient, propelling the repeated climbing–filling process that eventually forms a non-uniformly thick film of height *H*.

### 3.3. Macroscopic Capillary Wicking Dynamics

Different from the capillary rise in a closed tube [34] of a large diameter that enables a higher flux (larger amount of water raised) but results in a slower speed and lower height, capillary wicking in a mesoscopic trichomes array with different lengths but a fixed pitch and diameter exhibits a similar height change over time, as shown in Figure 4a, which implies that the interval filling process and substrate climbing process are independent. In other words, a faster wicking speed, higher wicking height, and thicker wicking film allow for the achievement of a higher flux at the same time.

In addition, capillary rise is usually classified into three regimes [35,36] (Appendix A): the initial stage is dominated by inertia following a scaling law, *H*~*t*; the final stage of thin-film wetting is influenced by viscosity, which follows Tanner’s law [37,38,39], *H*~*t*^1/10^; and the regime between the two stages is called the inertia–viscosity transition, which is described by the Lucas–Washburn equation as *H*~*t*^1/2^. However, mesoscopic structures enhance surface energy release and reduce the velocity gradient, which determine the viscous drag. The Reynolds number, *Re* = *ρue_n_*/*μ*, is *O*(10^2^), indicating that inertia dominates over viscosity, where *u* is the wicking velocity and *μ* is the viscosity of the liquid. The Bond number, Bo = *ρ*g*He_n_*/*σ*, is *O*(10^−2^), implying that the influence of gravity is negligible. Therefore, the governing equation for the mesoscopic wicking dynamics of a single interval column takes the following form:*d*(*mu*)/*d*t = *udm*/*dt* + *mdu*/*dt* = *F* = *σ***p*(8)
where *σ** is the equivalent surface tension considering the surface roughness [1,4] and *m* is the mass of the wicking liquid, which can be expressed as k*ρHpe_n_*, where k is a geometric factor and *u* = *H*′ is the first derivative of *H*. Considering that the thickness relates to the wicking height as *e_n_*~*βH* (Figure 3c), *m* is expressed as k′*ρH*^2^*p*, where k′ = k*β* and a different scaling law can be obtained from Equation (8) (see Appendix B for detail):(9)$$H\sim {\left(\frac{3\sigma^{*}}{2k^{\prime }\rho }t\right)}^{2/3}$$
which describes the transition regime dynamics of mesoscopic wicking and is in accord with the experimental results.

Furthermore, we confirm the general applicability of the new scaling law regarding the transition regime of capillary wicking on a mesoscopic trichomes array. The height change over time for different liquids wetting our fabricated materials is reported in Figure 4c, which reveals a similar trend, but the low surface tension of viscous liquids such as silicone oil exhibits a slower climbing rate compared to high-surface-tension liquids with a low viscosity, such as water. Dimensionless height, *H*/*l*_c_, with respect to normalized time, *σt*/*μ*, was used to assess deviations from the expected tendency of different liquids (Figure 4d). The results indicate that all liquids follow the scaling law at the beginning, but deviate from their expected tendencies in the late stage because the viscosity-negligible condition that *Re* is large does not hold for thin films of viscous liquids such as silicone oil and ethylene glycol in our experiments.

### 3.4. High-Flux Open Siphon Applications and Mass Transportation

Subsequently, we curved the straight plate into “n” shape (Figure 5a), so that the film on the trichomes array acts as a water path connecting two levels and facilitating water transfer from the higher level to the lower level, functioning as an open siphon device [40,41,42]. Compared to the “n” shape device without a trichomes array modification (Figure 5b), water spontaneously wicks between the trichomes once it touches the modified region and follows the trichomes’ path, flowing into the other side of the device. As water accumulates in the outlet side, the meniscus rises to touch the modified region, enabling continuous water transfer until the meniscus levels on the inlet side and outlet side are equal. Similar to a traditional siphon in a closed tube, the height difference between the inlet and outlet, *h*_out_ − *h*_in_, where *h*_in_ is the height difference from the inlet tank’s bottom to irs ridge top and *h*_out_ is the height difference from the outlet tank’s bottom to its ridge top, influences the siphon flux [43] (Figure 5c).

However, as reported in previous work, the flux of an open siphon exhibits a maximum flat stage [42], such that increases in the two-sided height difference have little influence on the flux. Instead, the maximum flux is influenced by the inlet height difference, *h*_in_, which determines the driving pressure difference across the free surface. The flux of our trichomes array achieved an open siphon, which is also strongly influenced by the *h*_in_, which the maximum flux reduces to half when the *h*_in_ increases from 1.00 cm to 1.25 cm. Except for the maximum flux, flux under small two-sided height differences is enhanced by capillarity if the *h*_in_ is small, which is attributed to a larger Laplace pressure difference across the meniscus for a small *h*_in_. Siphon applications that require high flux such as irrigation systems and drainage systems can be enhanced by adding a trichomes array with an optimized height difference on their surface.

The application of water wicking on a trichomes array extends beyond liquid transfer, which can be utilized for mass transportation as well. Various solutions carrying mineral salts or nutrients, such as glucose, wick on the biomimetic trichomes array and form a non-uniformly thick film (Figure 5d). The thinner film at higher levels evaporates first, causing solute crystallization to cover the surface. Over time, as the film gets thinner, solutes from the resources are transported to the surface at high levels (Figure 5e). In this way, a pitcher plant like *H*. *minor* can carry nutrients such as sugar and digestive liquid from the glandular organ at the bottom of pitcher to the upper pubescent zone, which attracts insects after crystallization. This process requires further investigation of the biological subject [31].

## 4. Conclusions

Inspired by the capillary wicking on the pubescent inside wall of *Heliamphora minor*, we fabricated a biomimetic array of millimeter-long and sub-millimeter-pitched trichomes through LCD-based 3D printing, with a high resolution of up to 18 μm, to investigate mesoscopic wicking dynamics. Our results demonstrate that the water wicking on a trichomes array consists of two independent processes: the filling intervals of the trichomes and the meniscus advancing on the substrate. The filling of the trichomes’ intervals is driven by capillary forces along the perimeter of the trichomes, resulting in the formation of a film with non-uniform thickness, and the film profile at each interval is described by a catenary line. Additionally, we unveiled that the transition regime of wicking on a mesoscopic-trichomes-modified surface is dominated by inertia. The dynamics follows a scaling law of *H*~*t*^2/3^, which is distinct from the *H*~*t*^1/2^ observed in traditional capillary rising or wicking on surfaces modified with microscopic structures. Because the thickness and wicking dynamics are independent, high flux and fast speed can be achieved at the same time, which can improve the flow efficiency and has potential in applications such as high-flux siphons. Finally, we examined the mass transportation inside the film and observed thin-film evaporation, resulting in solute crystallization at higher levels first, which indicates a method utilized by the plant to transport nutrients in the digestive fluid from the pitcher to the upper pubescent zone, assisting in the attraction of insects. This work, exploring the capillary wicking phenomena on mesoscopic structures, advances our understanding of high-flux wetting behaviors and improves the design of efficient liquid transfer devices.

## Figures and Tables

**Figure 1 biomimetics-09-00102-f001:**
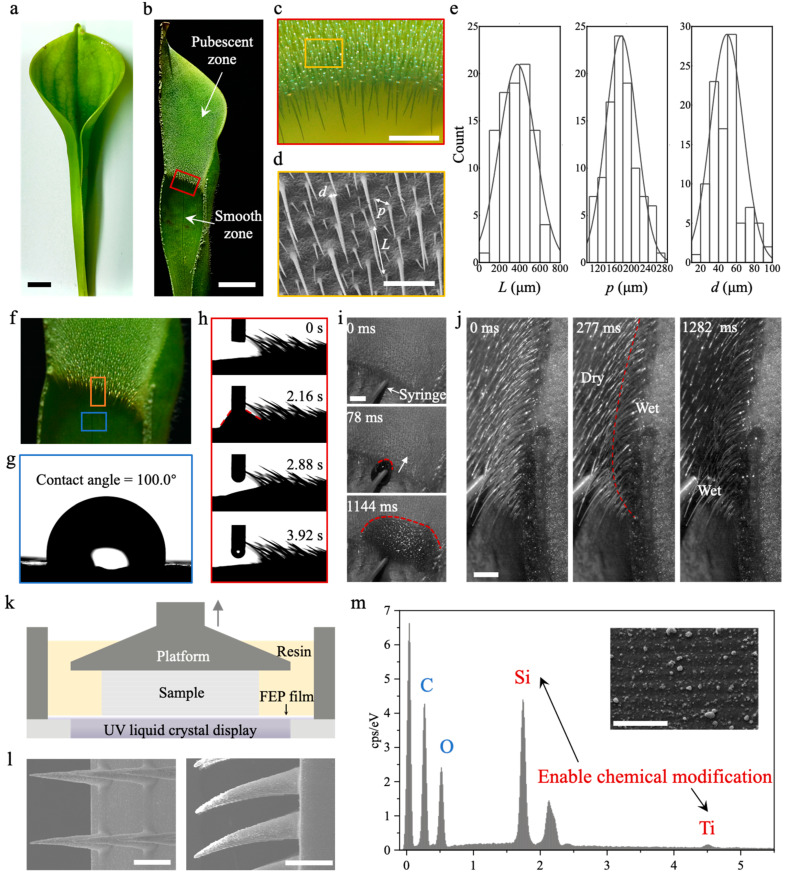
Characterization of *Heliamphora minor* and fabrication of biomimetic trichomes. (**a**) Optical images of *H. minor*; (**b**) sectional view of pitcher inside wall, above is a pubescent zone and below is a smooth zone; (**c**) stereomicroscope of the red-box-indicated region in (**b**); (**d**) SEM image of the yellow-box-indicated region in (**c**), trichomes of length *L* and base diameter *d* are arranged with a mutual pitch, *p*; (**e**) statistical analysis of *L*, *p*, *d* with normal distribution fitting; (**f**) illustration of sample resources from pitcher inside wall. The orange and blue boxes indicate the sample from the pubescent zone and smooth zone, respectively; (**g**) water contact angle on surface of smooth zone; (**h**) water contact angle changing on surface of pubescent zone, which gradually decreases to ~0°; (**i**) high-speed front view of water injected from syringe wicking on pubescent zone, the red dotted line indicates the fringe front; (**j**) high-speed side view of water wicking and filling the trichomes’ intervals, the red dotted line indicates the border of dry and wet regions; (**k**) schematic of bottom-up 3D printer based on photo-initiated polymerization of resin using liquid crystal display as UV source; (**l**) SEM images of 3D-printing fabricated biomimetic trichomes array, the left is top view and the right is side view; (**m**) energy-dispersive spectrometry of printed surface, Si and Ti elements are from SiO_2_ and TiO_2_ nanoparticles, which turn the surface superhydrophilic. The inset is a zoomed-in view of the surface. Scale bars: 1 cm in (**a**,**b**); 1 mm in (**c**,**d**,**i**,**j**); 500 µm in (**l**); 50 µm in (**m**).

**Figure 2 biomimetics-09-00102-f002:**
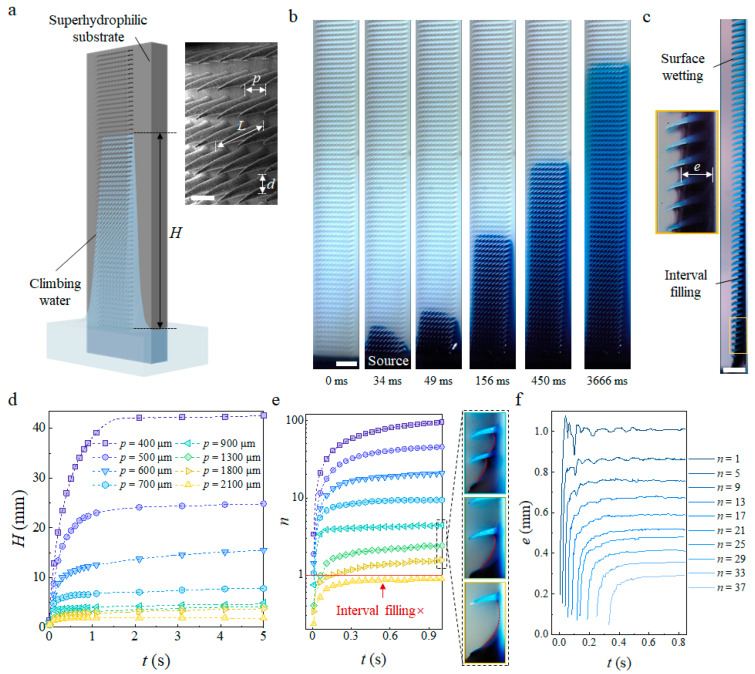
Capillary wicking on biomimetic trichomes array. (**a**) Schematic of capillary wicking on modified surface of superhydrophilic trichomes as the climbing water reaches a height of *H*. The SEM image shows the biomimetic trichomes array; (**b**) Oblique view of the wicking process on *L* = 2 mm, *p* = 500 µm, *d* = 200 µm trichomes array; (**c**) Side view of final wicking state. The film thickness, *e*, decreases with increasing *H*; (**d**) Climbing height, *H*, changes with time, *t*, on trichomes arrays of different pitches with fixed *L* = 2 mm and *d* = 200 µm; (**e**) Number of climbed intervals (*n*) changing with *t*. To the right are side views of *p* = 900 µm, 1300 µm, and 1800 µm from top to bottom; (**f**) Thickness of different intervals changing with *t* as the water climbs. Scale bars: 500 µm in (**a**), 2 mm in (**b**,**c**).

**Figure 3 biomimetics-09-00102-f003:**
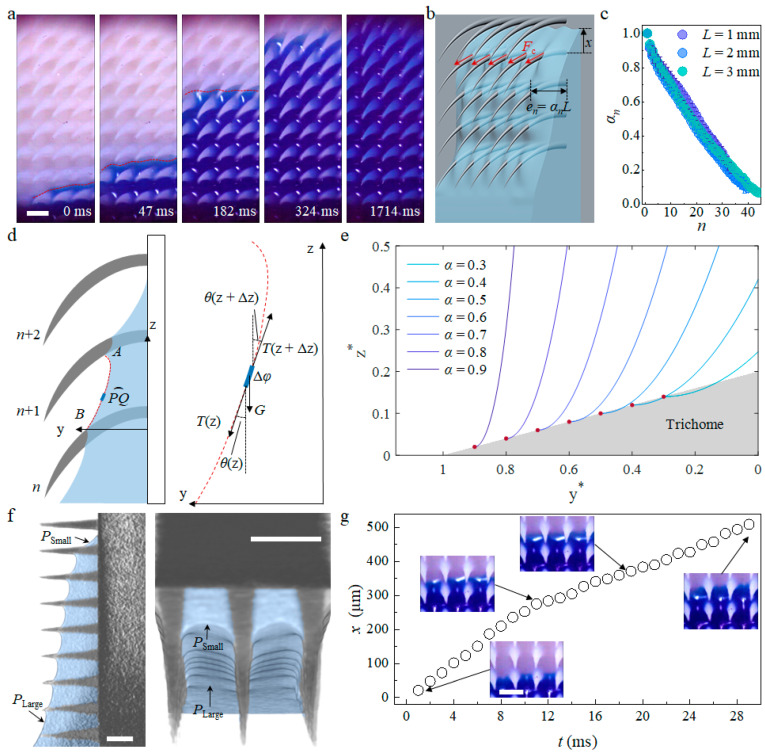
Microscopic capillary wicking dynamics and meniscus profile. (**a**) Microscopic view of capillary wicking on biomimetic trichomes array. The red dotted line indicates the meniscus front; (**b**) Schematic of capillary force on the perimeter of the trichomes and the film thickness in the *n*th interval. *e_n_* = *α_n_L*, where *α_n_* is the ratio between trichome thickness and length, and the meniscus front extends on the substrate to a distance of *x* from the (*n* + 1)th row; (**c**) Self-similarity of film profiles for trichomes arrays of different lengths. The ratio, *α_n_*, changing with *n,* collapses into a narrow region, implying a similar tendency across the arrays; (**d**) Catenary line schematic of meniscus *AB* in the *n*th interval (left), and force analysis of a small arc *PQ* in the meniscus of length Δ*φ* (right). The tension force, *T,* and gravity, *G,* are two major forces acting on two end points, and the tangential deflection angle is *θ*; (**e**) Meniscus profile of different trichome thickness to length ratios, *α*. Normalized z* = z/*p* and y* = y/*p* are used; (**f**) Film profile reconstructed from Micro-CT scanning. Both cross-sectional views in the y–z plane and perpendicular to z axis follow the catenary line, which results in a pressure gradient that is large at the bottom and small at the top. The blue false color indicates the water phase; (**f**) Meniscus extending with time, *t*, in a single interval at *H* ≈ 13 mm. The insets are in situ snapshots corresponding to the arrow-indicated point. Scale bars: 500 µm in (**a**,**f**,**g**).

**Figure 4 biomimetics-09-00102-f004:**
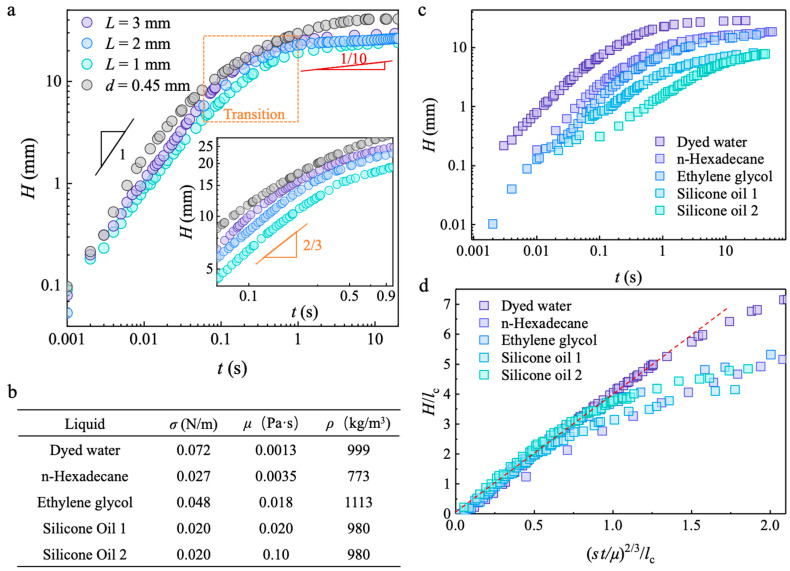
Macroscopic capillary wicking dynamics and generality of scaling law for different liquids. (**a**) Capillary wicking height, *H*, changes with time, *t*. The dynamics of capillary rise in a closed tube of diameter, *d* = 0.45 mm, is plotted in gray for comparison, and the logarithm coordinates are used to present the scaling law. The inset enlarges the transition regime between the inertia-dominated regime (*H*~*t*) and the viscosity-dominated thin-film spreading regime (*H*~*t*^1/10^). The scaling law fits the transition regime of capillary wicking that is *H*~*t*^2/3^ when the capillary is *H*~*t*^1/2^; (**b**) Properties of different liquids including surface tension, viscosity, and density; (**c**) Capillary wicking dynamics of different liquids in a trichomes array, with parameters of *L* = 2 mm, *p* = 500 μm, and *d* = 200 μm; (**d**) Dimensionless capillary wicking height with respect to normalized time. The trends of viscous liquids deviate from that of dyed water (indicated by the red dotted line) in the late stage due to the thin film and as the slow flow of the Reynolds number decreases, meaning viscous dissipation plays an important role in influencing the dynamics.

**Figure 5 biomimetics-09-00102-f005:**
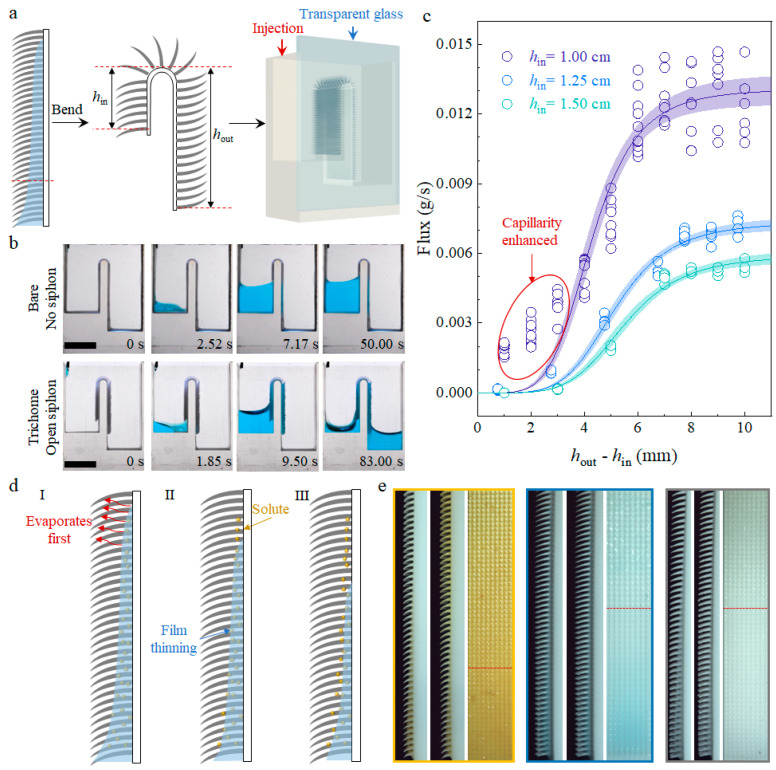
High-flux open siphon application and mass transportation through capillary wicking. (**a**) Schematic of open siphon construction, the straight surface is bent into an “n” shape with the inlet height difference *h*_in_ and outlet height difference *h*_out_, transparent glass was used for the observation of the water flow injected from inlet. The red dotted line to the left indicates the bending position; (**b**) Comparison between bare and trichomes-array-modified devices shows that no siphon forms on the bare side wall, while spontaneous wicking on the trichomes connects the two sides; (**c**) Flux measurement of different two-sided height differences for different *h*_in_. Meniscus deformation leads to large Laplace pressure, which enhances the flux, deviating from the fitting tendency; (**d**) Schematic of mass transportation and thin-film evaporation (I) leading to solute crystallization (II), (III) is the final state. (**e**) Side and front views of film thinning and solute crystallization. From left to right, the solutes are FeCl_3_, CuSO_4_, and glucose, respectively. Above the red dotted line is crystallized solute, which reflects light. Scale bars: 1 cm in (**b**).

## Data Availability

Data are contained within the article and Appendix A.

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
