# Peer review of "Capillary Wicking on Heliamphora minor-Mimicking Mesoscopic Trichomes Array"

_biomimetics, 2024, doi:10.3390/biomimetics9020102_

Round 1

Reviewer 1 Report

Comments and Suggestions for Authors

This paper reports a study of biologically inspired wicking, The study is specifically related to the species Heliamphora minor, and the experiments are set up to mimic the natural process.

The experiments have been carefully carried out and reported in thorough detail. 

The description of the work is not difficult to follow, but unfortunately the English is not fluent, and does not reflect the quality of the work.

Many symbols are used in the mathematical development.  It would be useful if a list of definitions was provided as an appendix to the paper.

All the physical behaviour described depends on the superhydrophilic surface of the trichome substrate.  This, as mentioned at the beginning of the paper, depends on the contact angle which in turn is determined by the intermolecular forces between water and the trichome substrate.  Clearly this is a vital parameter.  If the substrate has hydrophilic groups the contact angle will be low but will be large, or even greater than 90 if the substrate is hydrophobic.  The parameter is mentioned once at the beginning of section 2.1.  For a reader with more general interest, some additional discussion of the significance of this vital parameter would be a valuable addition.

A reference should be given for Tanner’s law.  Eg: Delgadino and Mellet; Communications in pure and applied mathematics, v74, p507 (2021).

Comments on the Quality of English Language

The description of the work is not difficult to follow, but unfortunately the English is not fluent, and does not reflect the quality of the work.

Reviewer 2 Report

Comments and Suggestions for Authors

The authors have successfully characterized multiscale capillary wicking on Heliamphora minor modified mimetic surface. The article has all the data needed but some additions and re-organization are suggested to improve the clarity of the contributions.

General Comments:

1) The introduction could benefit from more references related to wicking and biomimetic shape factors. 

2) Which high flux fluidic applications do you have in mind?

3) The effects of viscous drag are not discussed although silicone oils are used.  

4) Please remove porosity from the abstract since it has not been characterized.

4) Why is Figure 2 in the Materials? It is part of the Results. Related to this the supplementary material, Figures S1 and S2  should be moved into the main text as the it contains proof of surface modification.  Figure S2 is a good addition to the Introduction for making the case for the deviation from the Washburn equation.

Equations and Correlations:

1) The article mentions all the critical relevant correlations however they should be included namely the deviation from the Lucas Washburn equation illustrated in Figure 4, a major contribution from this work.

2) Although the results are clearly presented, a synthesis empirical equation or chart would be very useful to make the concepts more tangible.

Examples of such correlations are included in the following: (2019) 9:702 | DOI:10.1038/s41598-018-37368-y; https://www.nature.com/articles/s41467-021-23708-6

Comments on the Quality of English Language

Please consider English editing services.

Round 2

Reviewer 2 Report

Comments and Suggestions for Authors

Dear Authors,

Thank you for the addressing the comments  enhancing the high quality submission.